# Circular economy, environmental quality and tourism receipts in Europe: A time series data analysis

**Michael Odei Erdiaw-Kwasie**[1]*, **Kofi Kusi Owusu-Ansah**[2], **Matthew Abunyewah**[3], **Khorshed Alam**[4], **Abebe Hailemariam**[5], **Patrick Arhin**[6], **Kerstin K. Zander**[7], **Jonatan Lassa**[7]

**1** Business and Accounting Discipline, Faculty of Arts and Society, Charles Darwin University, Darwin, Northern Territory, Australia, **2** College of Arts and Sciences, University of Wyoming, Laramie, Wyoming, United States of America, **3** The Australasian Centre for Resilience Implementation for Sustainable Communities, College of Health and Human Sciences, Charles Darwin University, Darwin, Northern TerritoryAustralia, **4** Business School, University of Southern Queensland, Queensland, Australia, **5** Bankwest Curtin Economics Centre, Curtin University, Bentley, Western Australia, Australia, **6** Department of Spatial Planning, Dortmund University, Dortmund, Germany, **7** Northern Institute, Charles Darwin University, Darwin, Queensland, Australia

* michael.erdiaw-kwasie@cdu.edu.au

**Data Availability Statement:** All relevant data are within the paper and its Supporting information files.

## Abstract

The study examines how progress towards a circular economy (CE), patents related to recycling and secondary raw materials as a proxy for innovation, affect tourism receipts. The study uses Autoregressive Distributed Lag (ARDL) and Error Correction Method (ECM) to analyse time series data from EU countries from 2000 to 2020. Our estimates show that there exist long-run and short-run equilibrium relationships. In sum, evidence shows that promoting circular innovative practices, including recycling and using secondary raw materials in tourist destinations, could improve environmental quality and positively impact tourism receipts. The study concludes with policy and practical suggestions for circular economy innovation towards green tourism, destination management, and sustainable tourism.

## 1. Introduction

Tourism is one of the world's largest industries and, despite a hiatus in 2019/2020 because of a global pandemic, is growing at a fast rate, accounting for one-third of the world's total services trade [1]. The number of travellers worldwide grew rapidly from 25 million in 1950 to 1,466 million in 2019. Also, among the most significant trends in tourism is the growth in international tourism receipts (54%), which exceeds the growth in the world's GDP (44%) from 1950 to 2019 [1]. Approximately 1460 million international arrivals in 2019 generated $1,481 billion in international tourism receipts [2]. Furthermore, tourism contributes significantly to the economic growth of a destination by providing employment opportunities, accumulating foreign exchange earnings, and improving infrastructure [3–5]. In 2019 these figures represented more than 28% of the world's services exports [1]. Tourism has therefore been widely recognised as a catalyst for export trade and economic growth in many countries. However, despite the significant economic contributions of the tourism industry, top tourism destinations are

**Funding:** The authors received no specific funding for this work.

**Competing interests:** The authors have declared that no competing interests exist.

becoming increasingly concerned about the environmental impacts such as rising natural resource consumption, carbon emissions, litter, and pollution and the associated costs [3, 6, 7]. This paper investigates whether progress towards circular economy innovations in the form of recycling and the use of secondary raw materials improves tourism receipts.

Europe is the leading tourist destination in the world [8–10]. The region's attractiveness to tourists and main strengths in tourism is underpinned by the continent's growing cultural diversity, enviable infrastructural development, and the opportunity to travel within the Schengen zone with limited restrictions [11–13]. According to the World Bank [14], Germany, France, the United Kingdom, and Italy are the most preferred tourist destinations. They are ranked higher on the global travel and tourism competitiveness index. Over the past decades, the tourism industry in Europe has seen a continued expansion, despite occasional shocks such as the covid-19 pandemic outbreak. For instance, in 2018, Europe accounted for 51% and 36% of international tourist arrivals and global tourism receipts, respectively [2]. This is expected to increase rapidly in the next few decades due to the new tourism infrastructure, cultural and natural heritage transformation, and the rapid transition of Central-Eastern Europe economies into new tourism destinations [15, 16]. Regarding economic contribution, 10% of European Union countries' gross domestic products and 9% of the total workforce are generated from the tourism industry. In addition, one-tenth of Europe's non-financial business subsector are classified as tourism enterprises [17]. Hence, tourism's enormous contribution to Europe's socioeconomic development cannot be overlooked.

There is a concern, however, about tourism's impact on environmental degradation due to rising emissions [18–21]. For example, Lenzen et al. [19] shared that between 2009 and 2013, tourism's global carbon footprint increased from 3.9 to 4.5 GtCO2e, four times more than previously estimated. In a recent report jointly released by The World Tourism Organisation and the International Transport Forum, carbon dioxide emissions related to transport accounted for almost 5% of all manmade emissions and are projected to reach 5.3% or higher by 2030, primarily if tourism is not managed sustainably and ecologically to change the sector's production and consumption patterns [2]. According to the report, the sector's environmental impact is caused primarily by transportation and accommodation activities that utilise fossil fuels. In addition, tourism activities contribute to air pollution, noise, and waste generation. [22, 23]. The influx of visitors further exacerbates this problem. Likewise, Rico et al. [24] found that the average tourist's carbon footprint is between 43.0–111.6 kgCO2 eq/day, which is higher than the average value for a resident in Barcelona (5.8kgCO2 eq/day). Ritchie and Crouch [25], based on the tourism industry's failure to preserve nature for generations unborn, conclude that the industry is uncompetitive, even though it provides satisfying and memorable experiences. Thus, researchers and government officials call for ideal tourism approaches that balance socioeconomic development and environmental sustainability. The current tourism business model (take-make-dispose) heavily relies on easily accessible and cheap resources [26, 27].

Circular economy, a transformative and regenerative approach that restores and replaces the end-of-life of material, has been touted as a possible solution to mitigate energy, water and waste generation in the tourism industry [28–30]. In adopting circular economy principles in tourism [31, 32], non-renewable energy is shifted toward renewable energy, and many wastes are reduced or eliminated through the better use of materials, systems, and products. An eco-friendly approach to tourism is one of circular tourism's key elements, involving visitors, hosts, tour operators, and suppliers. According to the aforementioned arguments, although many studies have examined the circular economy, environmental quality, and economic returns linkage in general, not much attention has been paid to the tourism sector in particular. The motivation for our study is this gap. To fill this gap, this study examines the impact of

the circular economy on tourism receipts and environmental quality in the world's top tourist destinations across Europe. A deeper exploration and discussion of the relationship between circular economy, environmental quality, and tourism receipts are provided by this study. In light of this, this study aims to answer the following key research question: Does the circular economy affect tourism receipts in EU countries? Specifically, can circular economy patents associated with recycling and secondary raw materials significantly affect tourism receipts differently than traditional virgin materials-based destinations? If yes, then to what extent?

Our study contributes significantly in three ways. First, to the authors' knowledge, it is the first empirical study to introduce circular economy patent- related to recycling and secondary raw materials to tourism destination studies. The findings offer empirical evidence on the circular economy-environmental quality-tourism receipts relationship. This paper empirically explores this nexus by simultaneously including circular economy patent related to recycling and secondary raw materials as a proxy for innovation, environmental quality and tourism receipts in a multivariate framework, even though these concepts have been studied separately in various literature. Using a multivariate framework, we can investigate the entangled relationship and make up for the deficiency in the literature. Second, the literature review indicates that most empirical studies are based on panel data modelling. Even though panel data techniques provide efficient estimates, their conclusions and policy implications may not apply to individual countries due to heterogeneity [33, 34]. By utilising a time-series approach, this study offers policy guidelines for the world's top tourist destinations regarding the effects of circular economy on environmental quality and tourism receipts. Finally, the findings can assist policymakers in tourism-based EU countries in determining whether a circular economy positively impacts the country's tourism sector. In order to achieve sustainable tourism outcomes such as increased tourism revenues and safer environments, governments should formulate policies that encourage circular economy policies and strategies.

The paper proceeds as follows: the next section presents the study's literature review. Section 3 explains the method used in the study. Section 4 presents our results, and section 5 provides the discussion. Finally, section 6 presents the conclusion.

## 2. Literature review

### 2.1 Theoretical motivation for the study

Two theoretical models underpin this study: the destination image model [35] and the Value-Belief-Norm of environmentalism [36]. A destination image model demonstrates how negative events can affect tourists' choices of destinations [37, 38]. As per the destination image model, images formed by tourists represent an array of associations they mentally ascribe to. The theory postulates that tourists' information about a destination influences their decision to either visit or not visit a place. In addition to influencing tourists' travel decisions and behaviour towards a destination, the destination image model also impacts satisfaction levels and recollection of the experience that they have experienced. Recent studies have applied the destination image model within the tourism sector and assessed shared understanding of the tourism destination's internal issues [39–41]. According to Josiassen et al. [42], individuals keep memories of the tourism experiences of a destination and create destination imagery based on those experiences. This influences destination image formation, including satisfaction and visitation intentions. Based on Zahra [43] Bangladesh study, it appears that the tourist's image of the destination is shaped by the consistent flow of information between the destination and the home country. According to the study, most respondents have a negative perception of Bangladesh, which appears distorted and distorted. The adverse event experienced by visitors influences their perceptions about the destination and can negatively affect

their intentions to revisit and the recommendations they make to their social networks [44–46]. This will eventually reduce visitor numbers to these destinations. In our context, the negative experience could stem from pollution and litter, too much waste, the use of unsustainable materials, and non-existing recycling facilities, which make tourists aware of the unsustainable practices in their destinations.

This is where the Value-Belief-Norm framework is important. While the destination image model is concerned with the characteristics of the destination, the Value-Belief-Norm framework is tourist-based. The Value-Belief-Norm model has its roots in the social psychology discipline, and it stipulates that taking action with a pro-environmental intention is a normative moral obligation [36]. Recently, the Value-Belief-Norm model has gained traction with research on sustainable tourism and hospitality, focusing primarily on pro-sustainable tourism behavioural intentions [47, 48] and their likelihood of choosing green accommodations [49]. In Chen's [50] Taiwanese study, for instance, the causal sequence of variables in the VBN theory of pro-environmental behaviour is found to have direct and mediatory effects, such as awareness of consequences (AC), attributions of responsibility to self-beliefs, and personal norms (PN). According to Landon et al. [51] and Liu et al. [52], similar findings were also reported in the United States and Mongolia.

Based on the Value-Belief-Norm of environmentalism and the destination image model, this study examines the impacts of circular economy patents related to recycling, and secondary raw materials use on tourism receipts. Tourism destinations should strive for differentiation from their competitors by creating an attractive destination image, such as safety, eco-friendly, circular practices, and green travel initiatives, among many others. Although it is great to have more tourists unless these tourists are eco-tourists and careful about their consumption, the influx of tourists can be even worse for the environment. Tourist destinations with good ecological footprint image can increase tourism receipts and increase the number of tourists [53, 54]. Hence, we suggest that circular economy patents related to recycling and secondary raw materials used in tourism destinations can contribute to a positive image, which can impact tourist inflow and receipts.

## 2.2 Circular economy and tourism receipts

Circularity is essential in an increasingly sustainable world, but not every sustainability initiative supports circularity [55]. The term circular economy seems to have emerged more recently than sustainability. EMF [56] traces the circular economy to various schools of thought, including industrial ecology and cradle-to-cradle. However, sustainability is an older concept [57] and has been institutionalised by environmental movements and supranational organizations, particularly after the Brundtland report was published in 1987. There is no doubt that the circular economy seeks to achieve a closed loop by eliminating resource inputs, waste, and emission leakages [58, 59], but sustainability is a multifaceted concept, and different authors address a wide range of goals based on the agents considered and their interests [60]. Sustainable development and circularity have long been linked by visions, models, and theories.

As part of the UN Sustainable Development Goals (SDGs), responsible consumption and production (SDG12) are promoted due to their multiplier effects on societal growth and development [61, 62]. The circular economy can give companies opportunities to extend the economic lifespan of products [63, 64], minimise waste [65] and decrease their reliance on virgin resources [66]. In lieu of destroying value after use, a cycle of reusing, repairing, remanufacturing, or recycling is used to preserve value. Therefore, we need to develop new business models and innovative product designs that use non-toxic materials that can be endlessly recycled to achieve this goal [64, 67]. As we switch from our current consumption

methods to a circular economy, wealth and prosperity move from one system to another [68, 69]. By design, the system is regenerative, meeting all citizens' needs within the earth's natural capacity. Circular economy brings to light the damaging effects of economic activity on human and natural systems and then designs solutions to mitigate them. Taking circularity actions can help achieve Sustainable Cities and Communities (Goal 11), Responsible Production and Consumption (Goal 12), Climate Action (Goal 13), Life Below Waters (Goal 14), and Life on Lands (Goal 15) by reducing production, consumption, and waste, and encouraging community cooperation.

The tourism industry contributes significantly to CO2 emissions, and many tourists demand luxury accommodations and resource-intensive experiences [70]. Tourism is estimated to contribute 8% of global CO2 emissions, 72% of which are attributable to fuel combustion and land use changes. Manniche et al. [17] point out that tourism typically relies on cheap and easily accessible resources, produces solid waste, contributes to wastewater, and causes other environmental problems. Tourism can embrace a sustainable and resilient future by integrating circularity and advancing resource efficiency in its value chain [70, 71]. The circular economy offers tourism destinations the opportunity to maximize tourism's sustainable development impacts, creating more jobs and more inclusive local value chains, thus creating a virtuous circle between businesses and territories. This creates a positive impact on the local population.

Despite the growing recognition of the circular model contributing to sustainability, its application has over the years been centred on the manufacturing and construction industries [72, 73], with less emphasis on the service sector, particularly the tourism industry [74] even though it is an industry predominantly configured around the linear economy model [17, 75]. For example, a study by Pablo-Romero and colleagues [76] in the hospitality sector revealed a positive relationship between electricity consumption and overnight stays in Spain. A situation that makes the tourism sector a key threat to environmental sustainability.

The United Nations World Tourism Organization [2] has reported a global increase in international tourist arrivals from 850 million in 2008 to 1435 million in 2018, along with a linear consumption model common to the tourism industry. The literature on circular economy and tourism receipts has mainly focused on the possibility of using the latter to address the sector's many challenges [17]. Among these are reducing excessive food, water, and energy consumption and reducing waste and pollution. Although extensive research has been conducted on how a circular economy can address externalities within the wider tourism industry, little is known about how it impacts tourism receipts. According to research, many factors affect tourism receipts. Various factors, such as political stability, corruption, religious tensions, and the environment, contribute to tourism receipts [77]. Several studies, such as Butler [78] and Prayag et al. [79], examined the relationship between environmental quality and tourism in various countries. Environmental quality positively impacts tourism, and international tourists are more likely to visit these destinations.

A similar study by Tugcu and Topcu [80] identified that tourists are becoming more concerned about the effect of actions on the environment and are interested in choosing destinations where the circular economy model is incorporated into their consumption pattern. As a result of the growing public awareness that climate change and other severe environmental problems need to be addressed through change practices, a growing segment of tourists is actively seeking sustainable destinations, accommodations, and holiday experiences [81, 82]. Some studies indicate, however, that environmental concerns affect behaviours at home more than on vacation [83]. Also, according to Doran et al. [84], people with environmental value orientations are likelier to choose environmentally friendly destinations when they travel.

## 3. Methodology

### 3.1 Autoregressive Distributed Lag (ARDL) model

This study employs the ARDL Model and the Error Correction Method (ECM) to examine the short-run and long-run relationship between circular economy and tourism receipts, respectively.

To investigate the relationship between tourism receipts (TR) and circular economy (CE) in each country, we estimate an ARDL model of the form: As noted by Pesaran et al. [84], the model's flexibility makes it possible to integrate variables in a different order. From ARDL, one can construct a dynamic error correction model using linear transformation [85]. Furthermore, a distributional model with autoregressive distributed lags is superior regardless of whether the sample size is small or finite [86]. Furthermore, Pesaran et al. [84] demonstrate that modelling ARDL with appropriate lags will address berial correlation and endogeneity problems. Finally, using the ARDL model simultaneously provides unbiased and long-run cointegration estimates (Pesaran et al., 2001).

$$\Delta TR_t = \gamma_0 + \gamma_1 TR_{t-1} + \gamma_2 CE_{t-1} + \theta_1 \sum_{i=1}^{m} \Delta TR_{t-1} + \theta_2 \sum_{j=1}^{n} \Delta CE_{t-1} + \varepsilon_t \qquad (1)$$

Where TR is tourism receipt in billions of US\$, CE denotes circular economy proxied by the number of patents related to recycling and secondary raw materials. The parameters $\gamma_1$ and $\gamma_2$ are the long-run coefficients whose sum is equivalent to the error correction term at the VECM model, while $\theta_1$ and $\theta_2$ are short-run coefficients, and $\varepsilon_t$ is the error term.

To explore the long-run relationship between TR and CE, the null hypothesis of "no long-run relationship" is tested using an F-test of the joint significance of the coefficients on the lagged variables. Specifically, we tested the null hypothesis:

$H_0$: $\gamma_1 = \gamma_2 = 0$ (Cointegration is absent) against the alternative, $H_a$: $\gamma_1 \neq \gamma_2 \neq 0$ (Cointegration is present)

By applying the F-test, we examine and compare the 'no cointegration' null hypothesis against the alternative hypothesis stating the existence of cointegration. As noted in Pesaran [84], the F-statistic has non-standard distribution irrespective of whether the regressors are integrated of order 0, I(0) or order 1, I(1). The tabulated lower and upper-bound critical values are used to determine whether the null hypothesis has to be rejected. More precisely, the null hypothesis is rejected if the computed F-statistic is greater than the upper bound, suggesting a long-run relationship. In contrast, the null of no cointegration cannot be rejected if the F-statistic is smaller than the lower bound. If the F-stat falls between the lower and upper bound critical values, the result is inconclusive.

### 3.2 Granger causality

Time series analysis aims to determine whether changes in one variable cause changes in another and, if so, in which direction the causality will be (unidirectional, bidirectional, none). Granger [87] proposes that causality can be tested with F-tests which can be applied to examine whether lagged values of a variable Y provide statistically significant information about variable X when lagged X values exist. In this case, Y is not Granger-caused by X. Granger [87] described causality has both short-run and long-run components. A Wald test is used to determine short-run causality in error correction models. In order to perform the Granger causality test, we use the VAR model, which is the block exogeneity of Wald test. A VAR model captures the progression and interdependence of multiple time series. The asymmetrical approach is achieved by including a formula for each variable in the system. This explains its evolution

concerning its lags and the lags of other variables. VAR models describe variables as linear functions of their past changes over time. When testing Granger causality, the null hypothesis is that the independent variables cannot cause the dependent variable, whereas the alternative hypothesis is that the independent variables do cause the dependent variable. In our model, the null hypothesis is that CE does not lead to TR.

### 3.3 Data

We obtain data on tourism receipts from the World Development Indicators database of World Bank, while data on the circular economy is sourced from the Eurostat's database. Competitiveness and innovation in a circular economy are proxied by the number of patents related to recycling and secondary raw materials, while tourism receipts are expressed in billions of United States Dollars (US$). Data on these variables are drawn from 14 EU countries for the period spanning from 2000 to 2020. Among the sample countries included in this study, France, Spain, the United Kingdom, Austria and Germany ranked among the most visited tourist destination in Europe and the world. The choice of the time span from 2000–2020 for this study is mainly dependent on the availability of data on the circular economy. Table 1

**Table 1. Summary statistics of variables.**

|  | Variables | Mean | Standard Deviation | Maximum | Minimum |
|---|---|---|---|---|---|
| Germany | CE | 97.42 | 18.88 | 107.04 | 76.12 |
|  | TR | 46.18 | 11.63 | 59.45 | 24.18 |
| France | CE | 39.00 | 8.29 | 69.97 | 32.44 |
|  | TR | 49.53 | 10.37 | 65.36 | 32.19 |
| Spain | CE | 20.00 | 7.00 | 32.00 | 6.90 |
|  | TR | 57.33 | 17.09 | 81.42 | 18.35 |
| United Kingdom | CE | 21.16 | 4.61 | 32.47 | 16.00 |
|  | TR | 37.80 | 11.17 | 52.66 | 19.10 |
| Austria | CE | 10.35 | 3.48 | 17.92 | 3.86 |
|  | TR | 17.51 | 4.01 | 23.23 | 9.90 |
| Belgium | CE | 9.40 | 3.99 | 15.65 | 3.99 |
|  | TR | 10.97 | 2.34 | 15.25 | 0.00 |
| Russia | CE | 41.20 | 21.42 | 84.11 | 2.99 |
|  | TR | 12.17 | 5.31 | 20.20 | 4.72 |
| Poland | CE | 28.01 | 21.04 | 65.94 | 5.20 |
|  | TR | 10.00 | 3.47 | 15.71 | 4.73 |
| Ireland | CE | 2.44 | 1.31 | 4.61 | 0.50 |
|  | TR | 8.26 | 3.59 | 15.28 | 3.18 |
| Denmark | CE | 4.56 | 2.39 | 9.30 | 0.33 |
|  | TR | 6.00 | 1.44 | 9.10 | 3.67 |
| Portugal | CE | 2.49 | 2.18 | 5.60 | 0.00 |
|  | TR | 13.36 | 5.70 | 24.59 | 6.03 |
| Finland | CE | 11.67 | 3.64 | 17.03 | 5.63 |
|  | TR | 4.08 | 1.36 | 5.94 | 2.04 |
| Czech Republic | CE | 12.23 | 7.41 | 36.63 | 3.19 |
|  | TR | 6.75 | 1.93 | 9.23 | 3.34 |
| Romania | CE | 2.68 | 1.80 | 5.44 | 0.00 |
|  | TR | 1.84 | 1.16 | 4.24 | 0.31 |

Notes: Tourist receipt (TR) is in billions of current US$, while CE is measured in counts of patents related to a circular economy.

**Table 2. Model diagnostic test results.**

| Country | RESET test | Lagrange multiplier (LM) Test | Jarque-Bera normality Test | Breusch-Pagan Test |
|---|---|---|---|---|
| Germany | 0.92 | 0.46 | 0.34 | 0.74 |
| France | 0.115 | 0.091 | 0.46 | 0.11 |
| Spain | 0.082 | 0.695 | 0.56 | 0.07 |
| Austria | 0.85 | 0.82 | 0.92 | 0.055 |
| United Kingdom | 0.062 | 0.42 | 0.32 | 0.058 |
| Belgium | 0.72 | 0.93 | 0.54 | 0.097 |
| Russia | 0.70 | 0.32 | 0.73 | 0.77 |
| Poland | 0.064 | 0.83 | 0.88 | 0.43 |
| Ireland | 0.85 | 0.086 | 0.63 | 0.97 |
| Denmark | 0.33 | 0.33 | 0.89 | 0.28 |
| Portugal | 0.82 | 0.51 | 0.47 | 0.51 |
| Finland | 0.87 | 0.71 | 0.66 | 0.067 |
| Czech Republic | 0.82 | 0.15 | 0.70 | 0.66 |
| Romania | 0.105 | 0.66 | 0.40 | 0.085 |

Notes: All values reported are p-values. The RESET test is a Ramsey model specification test to check model stability. The Lagrange Multiplier suggests there is no issue of serial correlation, while the Jarque-Bera and Breusch Pagan tests confirm the normality and absence of heteroscedasticity, respectively.

provides the summary statistics of TR and CE for each country. The table shows that there are significant variations in the two variables across countries.

## 4. Results

### 4.1 Model development

The diagnostic statistics presented in Table 2 pertain to each county's ARDL model. Models were specified adequately according to the results of the Ramsey RESET tests. Based on the Lagrange multiplier (LM) test results, no serial correlation can be seen at the 5% significance level. The result of the Jarque-Bera test for normality was also normal. Further, our models did not exhibit heteroscedasticity according to the Breusch-Pagan test.

Next, we pre-tested for the stationarity property of the series, as it is a prerequisite for cointegration analysis to avoid spurious regression results. An Augmented Dickey-Fuller (ADF) test was used to determine whether a series has a unit root. The series would appear to have a unit root if the ADF tests fail to reject the null hypothesis. However, if the tests reject the null hypothesis, then the series has no root and is stationary. Table 3 presents the results of the unit root tests.

The results showed that TR in Ireland, Portugal, France, and Finland was not stationary at levels but stationary at their first difference, whereas CE in these countries was stationary at levels, indicating that their p-values were less than 5% significant and the absolute value of their test statistic was greater than 3. CE and TR in Germany, Austria, Russia, Denmark, the United Kingdom, Belgium, the Czech Republic, and Romania, on the other hand, were stationary at their first difference, thus order one I (1).

### 4.2 Long-run relationship

We begin with examining the long-run relationship between TR and CE using the ARDL bound test for cointegration. Table 4 presents the bound test results for each country in our sample.

**Table 3. Stationary test results.**

| | Variable | 5% Critical value | Test Stat I(0) | Test Stat I(1) | Decision |
|---|---|---|---|---|---|
| *Spain* | TR | -3.00 | -1.63 | -3.06 | I(1) |
| | CE | -3.00 | -2.84 | -6.11 | I(1) |
| *Germany* | TR | -3.00 | -0.71 | -4.49 | I(1) |
| | CE | -3.00 | -1.68 | -6.64 | I(1) |
| *Austria* | TR | -3.00 | -1.43 | -3.09 | I(1) |
| | CE | -3.00 | -2.87 | -6.56 | I(1) |
| *Russia* | TR | -3.00 | -1.63 | -3.29 | I(1) |
| | CE | -3.00 | -1.58 | -3.64 | I(1) |
| *Ireland* | TR | -3.00 | -1.86 | -3.33 | I(1) |
| | CE | -3.00 | -3.20 | | I(0) |
| *Denmark* | TR | -3.00 | -1.86 | -5.18 | I(1) |
| | CE | -3.00 | -2.35 | -4.08 | I(1) |
| *Portugal* | TR | -3.00 | -1.60 | -3.24 | I(1) |
| | CE | -3.00 | -3.31 | | I(0) |
| *France* | TR | -3.00 | -1.90 | -3.30 | I(1) |
| | CE | -3.00 | -3.39 | | I(0) |
| *United Kingdom* | TR | -3.00 | -1.60 | 3.01 | I(1) |
| | CE | -3.00 | -2.87 | -5.72 | I(1) |
| *Belgium* | TR | -3.00 | -1.48 | -3.81 | I(1) |
| | CE | -3.00 | -2.64 | -6.16 | I(1) |
| *Poland* | TR | -3.00 | -2.42 | -3.11 | I(1) |
| | CE | -3.00 | -1.87 | -4.19 | I(1) |
| *Finland* | TR | -3.00 | -2.77 | -3.24 | I(1) |
| | CE | -3.00 | -3.65 | | I(0) |
| *Czech Republic* | TR | -3.00 | -1.32 | -3.24 | I(1) |
| | CE | -3.00 | -1.57 | -6.68 | I(1) |
| *Romania* | TR | -3.00 | -2.16 | -3.91 | I(1) |
| | CE | -3.00 | -1.57 | -5.86 | I(1) |

The bound test showed that all countries, apart from Denmark and Finland had a computed F-statistic being less than the 5% critical value of the upper bound and lower bound values of 5.73 and 4.94, respectively. This means that the null hypothesis of no cointegration is not rejected. Denmark and Finland were the only countries where the long-run relationship between the TR and CE is evident. Table 5 reports the estimates of the long-run ARDL coefficients for these countries.

Progress towards a circular economy through promoting patents for recycling and secondary raw materials significantly impacted tourism revenue in Denmark and Finland over the long term, with a statistical significance of 5% (Table 5). This implies that a circular economy based on competitiveness and innovation can boost tourism receipts in Denmark and Finland. More precisely, the estimates showed that, all other things equal, a percentage increase in recycling and secondary raw materials leads to a rise in tourism receipts of 1.75% in Denmark and 2.8% in Finland in the long run.

### 4.3 Error correction model short run dynamic estimation

Following the analysis of the long-term relationship specified, an additional investigation was conducted to examine the short-term dynamics using the ECM model, as it is a suitable tool

**Table 4. ARDL bound test for cointegration.**

| Country | F-stat | Lower bound | Upper bound | Remarks |
|---|---|---|---|---|
| Germany | 3.195 | 4.94 | 5.73 | No Cointegration |
| France | 1.11 | 4.94 | 5.73 | No Cointegration |
| Spain | 4.73 | 4.94 | 5.73 | No Cointegration |
| United Kingdom | 3.195 | 4.94 | 5.73 | No Cointegration |
| Austria | 1.109 | 4.94 | 5.73 | No Cointegration |
| Belgium | 0.017 | 4.94 | 5.73 | No Cointegration |
| Russia | 0.114 | 4.94 | 5.73 | No Cointegration |
| Poland | 2.496 | 4.94 | 5.73 | No Cointegration |
| Ireland | 0.11 | 4.94 | 5.73 | No Cointegration |
| Denmark | 6.95* | 4.94 | 5.73 | Cointegration |
| Portugal | 0.028 | 4.94 | 5.73 | No Cointegration |
| Finland | 6.12* | 4.94 | 5.73 | Cointegration |
| Czech Republic | 0.895 | 4.94 | 5.73 | No Cointegration |
| Romania | 2.56 | 4.94 | 5.73 | No Cointegration |
| Hungary | 3.21 | 4.94 | 5.73 | No Cointegration |

Note:

* indicates rejection of the null hypothesis of no cointegration.

**Table 5. Estimated long-run ARDL coefficients.**

| Denmark | Variable | Coefficients | Standard Error | T-ratio | P-value |
|---|---|---|---|---|---|
| | (CE) | 1.75 | 0.47 | 3.72 | 0.04** |
| | *Number of obs. 17* | *R-Squared 0.71* | Adjusted R Squared 0.57 | | |
| Finland | **Variable** | **Coefficients** | **Standard Error** | **T-ratio** | **P-value** |
| | (CE) | 2.8 | 0.55 | 2.27 | 0.04** |
| | *Number of obs. 17* | *R-Squared 0.55* | Adjusted R Square 0.36 | | |

Note:

** denotes significant at 5%

for modelling the short-run dynamics of non-stationary but cointegrated variables, and it provides a framework for estimating the speed of adjustment towards the long-run equilibrium in order to obtain a reliable estimate of the short-term relationship between the dependent and independent variables. To obtain the short-term relationship, an estimation was made of the correlation between the dependent and independent variable. The coefficient of the first differenced variable was used to determine the short-term effects (Table 6).

$$\Delta TR_t = \gamma_0 + \sum_{i=1}^{m} \Delta TR_{t-1} + \theta_2 \sum_{j=1}^{n} \Delta CE_{t-1} + \varepsilon_t \qquad (2)$$

The $\varepsilon_t$ variable represents the error correction term which is the speed of adjustment towards the long-run equilibrium, having one period of shock in the model. A stable model error correction term should satisfy two important properties, proposed by Pahlavani et al. [88], which are negative and statistically significant at the 1% level of significance.

**Table 6. Estimated short-run ECM coefficients of CE.**

| | Variable | Coefficients | Standard Error | T-Ratio | P-value |
|---|---|---|---|---|---|
| *Spain* | D(CE) | 1.70 | 5.00 | 3.40 | 0.005** |
| | D(CE(-1)) | 1.01 | 7.03 | 1.43 | 0.177 |
| *Germany* | D(CE) | 1.12 | 0.59 | 2.15 | 0.048* |
| | D(CE(-1)) | -1.18 | 5.22 | -2.40 | -0.029 |
| *Austria* | D(CE) | -1.23 | 2.12 | 0.57 | 0.571 |
| | D(CE(-1)) | -1.31 | 2.10 | 0.63 | 0.541 |
| *Russia* | D(CE) | 3.18 | 2.12 | 0.47 | 0.645 |
| | D(CE(-1)) | -2.54 | 2.10 | -0.39 | 0.704 |
| *Ireland* | D(CE) | 4.39 | 3.10 | 0.91 | 0.375 |
| | D(CE(-1)) | -1.73 | 3.12 | -0.31 | 0.763 |
| *Denmark* | D(CE) | 4.37 | 1.72 | 2.54 | 0.026* |
| | D(CE(-1)) | 4.46 | 1.89 | 2.35 | 0.040* |
| *Portugal* | D(CE) | 1.30 | 0.63 | 2.16 | 0.049* |
| | D(CE(-1)) | 1.98 | 0.62 | 0.62 | 0.55 |
| *France* | CE | 2.98 | 2.76 | 1.08 | 0.298 |
| | CE(-1) | 1.54 | 2.87 | 0.54 | 0.600 |
| *United Kingdom* | D(CE) | 8.85 | 4.69 | 2.05 | 0.049* |
| | D(CE(-1)) | -2.87 | 4.78 | -0.60 | 0.557 |
| *Belgium* | D(CE) | 1.62 | 1.15 | 2.03 | 0.048* |
| | D(CE(-1)) | 1.03 | 1.57 | 0.65 | 0.527 |
| *Poland* | D(CE) | 6.51 | 2.02 | 2.20 | 0.044* |
| | D(CE(-1)) | 7.15 | 2.30 | 1.22 | 0.210 |
| *Finland* | D(CE) | 1.63 | 0.42 | 2.62 | 0.023* |
| | D(CE(-1)) | 1.44 | 0.42 | 1.66 | 0.124 |
| *Czech Republic* | D(CE) | -2.30 | 0.44 | -0.55 | 0.59 |
| | D(CE(-1)) | -1.06 | 0.42 | -0.33 | 0.745 |
| *Romania* | D(CE) | 1.20 | 1.50 | 0.85 | 0.410 |
| | D(CE(-1)) | 1.18 | 2.20 | 0.54 | 0.598 |

Notes:

**, and * denote significance at 1% and 5% levels, respectively.

Regarding the cointegration results in Table 4, the error correction term (εt) of the specified model is statistically significant at the 1% significance level and negative in Denmark and Finland. This indicates that any deviation from the long-run equilibrium will be corrected in the next period, with speeds of adjustment estimated to be 43% for Denmark and 44% for Finland. In other words, shocks to the system are absorbed relatively quickly, and the model adjusts towards its long-run equilibrium relatively fast in Denmark and Finland.

The results showed that patents relating to recycling and secondary raw materials significantly impact tourism revenue in Spain at a 1% level of significance (Table 6). This suggests that Spain may experience higher tourism receipts if there is increased competitiveness and innovation in the circular economy. Specifically, a percentage increase in patents relating to recycling and secondary raw materials would lead to a 1.70 percent increase in tourism receipts in Spain in the short run, assuming all other things remain the same. Therefore, competitiveness and innovation inherent in circular practices are important short-term determinants of tourism receipts in Spain.

An increase in patents related to recycling and secondary raw materials was associated with a significant increase in tourism receipts in Germany, the United Kingdom, Belgium, Poland, Finland, Denmark, and Portugal within a short period of time (Table 6). Furthermore, tourism receipts in these countries are heavily influenced by a patent related to recycling and secondary raw materials, at a significance level of 5%. The findings of this study confirm the argument that innovation and competitiveness are essential factors that influence tourism receipts in these countries.

In the short run, a positive but insignificant relationship was found between TR and CE in France, The Czech Republic, Austria, France and Romania. This means that an increase in patents pertaining to the recycling and secondary materials will not result in a significant rise in tourism receipt in these countries, all other things being equal. In conclusion, progress toward a circular economy by increasing patents relating to secondary raw materials and recycling leads to an increase in tourism receipts in most of the countries in the sample, suggesting the benefits of a transition to a circular economy.

## 4.4 Results of granger causality

After obtaining the short-term relationship between CE and TR, we perform the ECM, the Granger Causality tests, to ascertain if the explanatory variables affect the dependent variable in the short run. Table 7 displays the results of the Granger Causality test.

The results in Table 7 show that there is a short-run causal relationship between patents related to recycling and secondary raw materials and tourism receipts in Spain, Germany, the United Kingdom, Belgium, Denmark, Portugal and the Czech Republic. Therefore, a short-run unidirectional Granger causality runs from CE to TR, suggesting that competitiveness and innovation in a circular economy are associated with higher tourism receipts in these countries. In Romania, the circular economy is also impacted by tourism receipts. Meanwhile, there was no unidirectional or bidirectional causality relationship between these patents and tourism receipts in France, Austria, Ireland, and Finland.

**Table 7. Granger causality wald test.**

| Country | Test Statistic | Prob. CE→TR | Prob. TR→CE |
|---|---|---|---|
| Spain | 2.38 | 0.034* | 0.0583 |
| France | 4.527 | 0.104 | 0.553 |
| Germany | 1.31 | 0.045* | 0.099 |
| United Kingdom | 4.2 | 0.025* | 0.055 |
| Austria | 2.24 | 0.326 | 0.877 |
| Belgium | 8.56 | 0.014* | 0.334 |
| Russia | 0.43 | 0.81 | 0.91 |
| Poland | 0.37 | 0.829 | 0.511 |
| Ireland | 1.82 | 0.402 | 0.983 |
| Denmark | 8.17 | 0.032* | 0.84 |
| Portugal | 9.76 | 0.007* | 0.41 |
| Finland | 0.59 | 0.74 | 0.89 |
| Czech Republic | 5.37 | 0.035* | 0.376 |
| Romania | 15.5 | 0.97 | 0.004* |

Notes:

* denotes the existence of granger causality.

## 4.5 Model stability

In this study, we employed the Cumulative Sum of Recursive Residuals (CUSUM) and the Cumulative SUM of Recursive Residuals Squares (CUSUMSQ) initiated by Pesaran and Pesaran [89] to test the robustness of the models. The graphical representation of the CUSUM and CUSUMSQ is shown in Fig 1. The guideline for using this approach indicates that the model parameters are stable and consistent if the plots fall within the 5% level of the critical bound. Our results show that the CUSUM and the CUSUMSQ fall within the boundaries of all countries except Germany and Romania throughout the period. However, we found that the estimates converge to zero in the long run, which means that the models are stable and consistent.

## 5. Discussion

Tourism has a significant economic contribution in terms of receipts, GDP and employment in many economies [1]. In the EU, tourism has developed exponentially, making it one of the

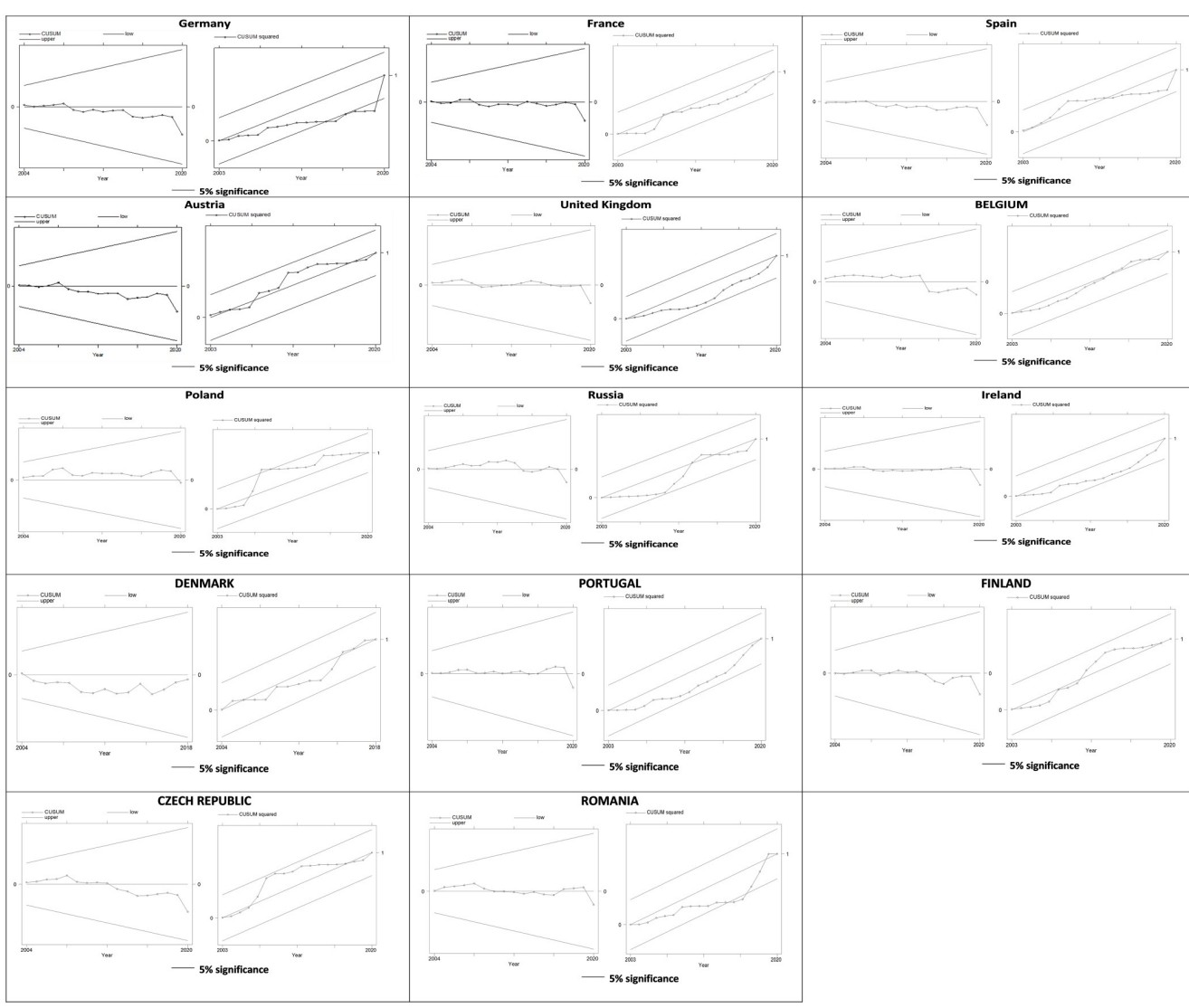

**Fig 1. Model and structure stability (CUSUM and CUSUMSQ) test.**

most dynamic forms of economic activity around the globe. France, Spain, Austria, Germany, and the United Kingdom are among the top ten most visited countries by international tourists and, consequently, the top ten countries with the highest travel receipts. However, despite the significant economic contributions of the tourism industry, top tourism destination countries are becoming increasingly concerned about environmental impacts such as rising natural resource consumption, carbon emissions, litter, and pollution [6, 7, 18, 19]. This paper investigates whether progress towards a circular economy improves tourism receipts for the host destination.

The ARDL, the Error Correction Model (ECM) and the Granger Causality tests were utilised to examine the relationship between circular economy and tourism receipt. The findings of the study show that competitiveness and innovation in a circular economy, measured by the number of patents related to recycling and secondary materials, have a significant positive effect on tourism receipts. On average, a percentage increase in patents on the recycling and secondary raw material use increases tourism receipts in Spain by 1.70%, Germany by 1.12%, the United Kingdom by 8.85%, Belgium by 6.51%, Denmark by 4.37%, Portugal by 1.3% Finland by 1.63% and Poland by 6.51% in the short run. In the long run, a percentage increase in patents related to recycling and secondary raw material leads to increased tourism receipts in Denmark and Finland by 1.75% and 2.8%, respectively. In sum, the results showed that the competitiveness and innovation in a circular economy drive tourism receipts across many EU countries, particularly the top five tourism destinations in Europe—France, Spain, Austria, Germany and the United Kingdom. Therefore, the findings demonstrate that circular economy policies can benefit the tourism industry.

Results indicate that circular economies are a crucial factor driving tourism receipts in most European countries. Others have observed similar results [90–93]. Furthermore, since EU countries are also global leaders in formulating environmentally friendly tourism policies, our findings demonstrate that the promotion of recycling and the use of secondary raw materials among EU countries significantly impact tourism revenues. Therefore, to ensure a smooth transition to a circular economy, the EU must create a competitive and sustainable tourism industry that is also low-carbon and resource-efficient.

Additionally, the Granger causality tests reveal a unidirectional causality running from TR to CE for most of the countries in the sample. The results suggest that progressing towards a circular economy causes an increase in tourism receipts in many EU countries rather than the other way around. From our empirical evidence, we can conclude that circular economies are important contributors to tourism receipts in most countries. These study findings confirm the earlier works of Ibn-Mohammed et al. [70] and Girard and Nocca [94]. Our empirical findings support a predictive relationship between circular economy and tourism receipts. In the sampled regions, circular practices seem related to tourism receipts since their results appear to be an important indicator. Thus, governments and policymakers in this region and the world's largest tourism markets should provide priority to predictive circular economy factors that enhance tourism receipts. Adding to this, to increase tourism revenues, circular economy initiatives can be promoted in tourism-based economies.

Apart from traditional methods for improving tourism receipts in host countries, this study also recommends that EU countries exploit the circular economy's potential to reduce the sector's environmental footprint and boost tourism receipts by leveraging circular economy-led innovation. By recycling and using secondary raw materials, circular economy-led solutions increase tourism receipts through two channels. First, recycling creates employment and wages- the material sorting, transfer, and transformation into new products can create opportunities for drivers, recycling and rubbish collectors, factory workers, and general and production managers [31, 94] Furthermore, government revenue is generated through taxation–

Waste and landfill levies have an important role in protecting the environment and generating substantial revenues for the states.

In tourism, companies investing significant capital into secondary raw materials should be recognised by levy discount percentages [95, 96]. According to this study, to maximise tourism receipts with minimal compromise on environmental quality and resource efficiency, EU countries should gradually move towards circular tourism, which can substantially improve tourism profitability through job creation, wage increases, and sustained tax policies. Finally, the shift to circular tourism can help redefine tourism destinations as assets consisting of natural and social stocks that must be protected and optimised for the long-term good of all stakeholders.

## 6. Conclusion and implications

The tourism industry is widely known to be a higher consumer of resources and a waste generator, including the unsustainable use of resources. The linear model of production and consumption prevalent in the tourism sector underpins the rising phenomenon. The global tourism industry has been concerned about how this could affect its growth, resilience and sustainability. However, no empirically tested evidence exists for how shifting to a circular economy can impact tourism receipts. Using a time series dataset from fourteen selected European countries from 2000 to 2020, we investigated the effects of a circular economy on tourism receipts.

Using the ARDL model, the Error Correction Model (ECM) and Granger Causality tests, we found that promoting the efficiency of recycling and secondary raw materials is positively associated with tourism receipts. Additionally, we checked for robustness and analysed patents by country. Then, we performed various robustness analyses considering the top European tourist destinations based on data availability and excluding outliers. Finally, we completed these robustness analyses and sensitivity tests based on our diagnostic and structural stability test results of the variables. Our results are robust to the various alternatives and sensitivity checks. In this context, increasing the use of recycling and secondary raw materials in tourist destinations can be an effective policy tool to promote tourism revenues. Furthermore, there is evidence that promoting circular practices in tourist destinations could increase international tourism by improving environmental quality and positively impacting tourism receipts. The results show that enhancing competitiveness and innovation in the circular economy is one way to drive circularity in the tourism industry across Europe.

The findings of this study have important policy implications. First, circular economy innovations in tourism should be accelerated. According to the empirical findings of this study, circular economy increases tourism receipts in Europe's top tourism destination. Tourism-based economies must urgently promote circular initiatives that are regenerative in nature in order to counter the impact of unsustainable tourism activities on the environment. To effectively deal with unsustainable tourism's degrading effects, policymakers need to implement and monitor a strict environmental framework. Also, the circular economy promotes environmental quality, and good ecological footprints lead to higher tourism receipts because they attract a large number of tourists. There is therefore a need for tourism-based countries to shift from over-reliant on environmentally damaging practices like single plastics, virgin materials, and increased waste generation to investing in circular economy initiatives like promoting reusable materials, recycling tourist waste, and utilizing secondary raw materials. By reducing environmental problems associated with unsustainable tourism across a wide range of tourist destinations, circular economy initiatives can help improve their overall revenue. It is worth also mentioning that there is a need for policymakers and practitioners to ensure that country-

specific heterogeneous effects are taken into consideration when implementing circularity-based tourism strategies. To achieve environmental sustainability in the tourism sector, policy-makers should use circular economy standards as part of the regulatory framework. In this aspect, as discussed previously, institutions play a significant role in implementing appropriate circular economy initiatives in the tourism sector in all regions. aspect.

As presented in this paper, a broader framework considering more than one circular indicator is recommended. Using such a broader framework, we can understand how tourism receipts and destination image are impacted by individual circular indicators from other streams. It is also worth exploring, though outside the scope of this paper, how tourism destinations and the industry can gain a competitive and innovative edge in emerging and developing countries by implementing the circular economy. Circular economy innovation may have spatial spillover effects between regions, and the spatial econometric model can be used to test the direct and indirect effects of circular economy innovation on regions. A further area of research is how to make circularity in tourism compatible with environmental pollution reduction to mitigate climate change.

## Supporting information

**S1 Dataset.**
(XLSX)

**S2 Dataset.**
(XLS)

**S1 File.**
(DOCX)

## Author Contributions

**Conceptualization:** Michael Odei Erdiaw-Kwasie, Matthew Abunyewah.

**Data curation:** Michael Odei Erdiaw-Kwasie, Kofi Kusi Owusu-Ansah, Matthew Abunyewah.

**Formal analysis:** Michael Odei Erdiaw-Kwasie, Kofi Kusi Owusu-Ansah, Khorshed Alam, Kerstin K. Zander, Jonatan Lassa.

**Methodology:** Michael Odei Erdiaw-Kwasie, Kofi Kusi Owusu-Ansah, Matthew Abunyewah, Khorshed Alam, Abebe Hailemariam, Patrick Arhin, Kerstin K. Zander.

**Resources:** Patrick Arhin, Jonatan Lassa.

**Supervision:** Michael Odei Erdiaw-Kwasie, Khorshed Alam.

**Visualization:** Kofi Kusi Owusu-Ansah, Abebe Hailemariam.

**Writing – original draft:** Michael Odei Erdiaw-Kwasie, Matthew Abunyewah.

**Writing – review & editing:** Michael Odei Erdiaw-Kwasie, Kofi Kusi Owusu-Ansah, Matthew Abunyewah, Khorshed Alam, Abebe Hailemariam, Patrick Arhin, Kerstin K. Zander, Jonatan Lassa.

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
