## [Decision Letter · Decision Letter 0]

27 Jan 2023

PONE-D-23-00742Circular economy and tourism receipts in Europe: A Panel data analysisPLOS ONE

Dear Dr. ERDIAW-KWASIE,

Thank you for submitting your manuscript to PLOS ONE. After careful consideration, we feel that it has merit but does not fully meet PLOS ONE’s publication criteria as it currently stands. Therefore, we invite you to submit a revised version of the manuscript that addresses the points raised during the review process.

We look forward to receiving your revised manuscript.

Kind regards,

Nikeel Nishkar Kumar

Academic Editor

PLOS ONE

Journal Requirements:

"No funding"

"No competing interest"

6. Please ensure that you include a title page within your main document. We do appreciate that you have a title page document uploaded as a separate file, however, as per our author guidelines (http://journals.plos.org/plosone/s/submission-guidelines#loc-title-page) we do require this to be part of the manuscript file itself and not uploaded separately.

Additional Editor Comments:

Please note formatting requirements for the journal. The revised paper should reflect this.

Reviewers' comments:

Reviewer's Responses to Questions

**Comments to the Author**

1. Is the manuscript technically sound, and do the data support the conclusions?

Reviewer #1: Yes

Reviewer #2: Yes

Reviewer #3: Partly

2. Has the statistical analysis been performed appropriately and rigorously? 

Reviewer #1: Yes

Reviewer #2: Yes

Reviewer #3: Yes

3. Have the authors made all data underlying the findings in their manuscript fully available?

Reviewer #1: No

Reviewer #2: Yes

Reviewer #3: No

4. Is the manuscript presented in an intelligible fashion and written in standard English?

Reviewer #1: Yes

Reviewer #2: Yes

Reviewer #3: Yes

5. Review Comments to the Author

Reviewer #1: 1. Sustainability and Circular Economy – There can be a discussion on this included in the Literature Review as circular economy practices may be deemed very similar to sustainable practices by readers. Therefore, a clear distinction needs to be made and the connection, for example, whether sustainable practices drive circular economy or vice-versa.

2. Page 5 – Stern et al, 1999 – capital “S”

3. Font colour – Black (Incertain places in seems that text is grey

4. Manuscript structure - align to that of the journal.

5. CUSUM squared issue for Germany and Romania.

6. Include limitations for the study.

7. Use Vancouver Style for referencing

Reviewer #2: I would like to sincerely thank the authors for submitting the article entitled: Circular economy and tourism receipts in Europe: A Panel data analysis. This study is exciting and authors have managed to gather insights using panel data and contribute towards the tourism literature. While the results are noteworthy there are limitations of this paper that may not meet the novelty of the journal currently. The following critics will be beneficial in reshaping the paper suitable for the journal in future.

1. I suggest the authors read the instructions for formatting the paper as per the requirement of the journal. (Font color varies in certain places and use of citation)

2. Justify your reasoning for employing the ARDL approach from tourism literature and perspective.

3. The "Value-Belief-Norm of environmentalism" theory lacks supporting evidence from country-specific studies. Provide literature specific to the model proposed and its relevance to tourism in some countries.

4. The second theoretical model incorporated is the destination image model. Please explore this angle further and complement the research contribution.

5. There is an overall gap between research findings and the linkage towards the paper's uniqueness.

6. Please be specific to which SDG Goal the present study is related; (egSDG1, 2, 3 ….), and highlight the particular goal in the abstract in some parts of the literature.

7.  Provide a citation for the statement that claims "Consequently, such an image can increase tourism

receipts and increase the number of tourists. However, although it is great to have more tourists

unless these tourists are eco-tourists and careful about their consumption, the influx of tourists

can be even worse for the environment."( Page 6)

8. Include a separate paragraph on Circular economy and Sustainable tourism.

9. Shorten long sentence into one sentence or split the sentence:  " Furthermore, the increasing public awareness that climate change and other severe

environmental problems must be addressed through change practices indicates that a growing

segment of tourists actively searches for destinations, accommodations, and holiday

experiences with a clear sustainability profile (Buffa, 2015; Doran et al., 2017), although some

studies indicate that environmental concerns translate stronger into behaviour at home than on

vacation (Mehmetoglu, 2010)."

10. Page 7, second paragraph, first line ; please rephrase with a meaningful outcome as it is unclear. "and the linear

consumption model endemic to the tourism industry, literature on circular economy and

tourism receipts has predominantly focused on how the former may assist with the many

challenges in the sector (Manniche et al., 2017)." Please rectify the phrase "how the former may assist with the many challenges in the sector" it is a clear indication of the passive sentence. Also, what does "the former" mean in the sentence?  

11. The methodology section is well articulated.  

12. How is the circular-led tourism hypothesis justified in the discussion section? Where has this been used previously?

13. It is suggested that discussion of results is well suited with theoretical consideration. Since there are two models being proposed in the literature but nothing has been mentioned in the discussion. I suggest linking the model being proposed in the literature and supporting with originality of this paper.

14. Please be mindful of the policy recommendation in the conclusion section. Justify your policy and its relevance to particular sectors.

15. Attached are the following works of literature that will be useful in replenishing the current study (a) Hernández, J. M., & León, C. J. (2013). Welfare and Environmental Degradation in a Tourism-Based Economy. Tourism Economics, 19(1), 5–35. https://doi.org/10.5367/te.2013.0191

(b) Kumar, N. N., Patel, A., Chandra, R. A., & Kumar, N. N. (2021). Publication bias and the tourism-led growth hypothesis. PloS one, 16(10), e0258730.

I wish the authors all the best in future.

Reviewer #3: Please note the following comments:

My comments relate mostly to the methodology of the study

Is it true that there are only 17 observations for the long run models for Denmark and Finland in Table 5? This is too less, even for ARDL models. Also, it is surprising that only 2 of the many countries considered are cointegrated. I find this very hard to believe. Because the data span is so less for each country, why not consider panel ARDL models such as the pooled mean group approach which is also developed by Pesaran, Shin and Smith (1999).

The results for Denmark and Finland are stable according to the CUSUM and CUSUMSQ plots. However, many of the other countries are not. This instability can possibly impact your long run results affecting the outcome of cointegration. It may indicate specification issues such as an incorrect functional form or the exclusion of structural breaks, however I do not know how you can possibly justify including structural breaks with a sample size of just 17! I then went to check the results of the RESET test in Table 2 and to my surprise, the notes to this table indicate that p-values are reported in parenthesis. I did not know that p-values could go up to 5.86. Please check before submitting.

Therefore, my recommendation is a major revision but the authors need to seriously address the methodological shortcomings of their paper.

6. PLOS authors have the option to publish the peer review history of their article (what does this mean?). If published, this will include your full peer review and any attached files.

Reviewer #1: No

Reviewer #2: No

Reviewer #3: No

---

## [Author Response · Author response to Decision Letter 0]

10 May 2023

Response file has been uploaded by the authors

---

## [Decision Letter · Decision Letter 1]

13 Jun 2023

PONE-D-23-00742R1Circular economy, environmental quality and tourism receipts in Europe: A time series data analysisPLOS ONE

Dear Dr. Michael,

Thank you for submitting your manuscript to PLOS ONE. After careful consideration, we feel that it has merit but does not fully meet PLOS ONE’s publication criteria as it currently stands. Therefore, we invite you to submit a revised version of the manuscript that addresses the points raised during the review process.

We look forward to receiving your revised manuscript.

Kind regards,

Nikeel Nishkar Kumar

Academic Editor

PLOS ONE

Journal Requirements:

Reviewers' comments:

Reviewer's Responses to Questions

**Comments to the Author**

1. If the authors have adequately addressed your comments raised in a previous round of review and you feel that this manuscript is now acceptable for publication, you may indicate that here to bypass the “Comments to the Author” section, enter your conflict of interest statement in the “Confidential to Editor” section, and submit your "Accept" recommendation.

Reviewer #1: All comments have been addressed

Reviewer #2: All comments have been addressed

2. Is the manuscript technically sound, and do the data support the conclusions?

Reviewer #1: Partly

Reviewer #2: Yes

3. Has the statistical analysis been performed appropriately and rigorously? 

Reviewer #1: I Don't Know

Reviewer #2: Yes

4. Have the authors made all data underlying the findings in their manuscript fully available?

Reviewer #1: Yes

Reviewer #2: Yes

5. Is the manuscript presented in an intelligible fashion and written in standard English?

Reviewer #1: Yes

Reviewer #2: Yes

6. Review Comments to the Author

Reviewer #1: All comments have been addressed.

However there are some additional coents based on the new version:

- on Page 17, the last paragrapgh start with although but there is no indication of although what. Please revise.

- On Page 4, 5th line. "patent patents". Please revise or rebut.

Reviewer #2: I would like to thank the authors for responding well towards the comments. I also hope that those comments will be beneficial in future prospects.

7. PLOS authors have the option to publish the peer review history of their article (what does this mean?). If published, this will include your full peer review and any attached files.

Reviewer #1: No

Reviewer #2: No

---

## [Author Response · Author response to Decision Letter 1]

17 Jun 2023

Response to reviewer file uploaded

---

## [Editor Report · Decision Letter 2]

19 Jun 2023

Circular economy, environmental quality and tourism receipts in Europe: A time series data analysis

PONE-D-23-00742R2

Dear Dr. Michael,

We’re pleased to inform you that your manuscript has been judged scientifically suitable for publication and will be formally accepted for publication once it meets all outstanding technical requirements.

Kind regards,

Nikeel Nishkar Kumar

Academic Editor

PLOS ONE
---

## [Editor Report · Acceptance letter]

6 Jul 2023

PONE-D-23-00742R2 

Circular economy, environmental quality and tourism receipts in Europe: A time series data analysis 

Dear Dr. Erdiaw-Kwasie:

I'm pleased to inform you that your manuscript has been deemed suitable for publication in PLOS ONE. Congratulations! Your manuscript is now with our production department. 

Kind regards, 

on behalf of

Dr. Nikeel Nishkar Kumar 

Academic Editor

PLOS ONE